# The Search for Suitable Habitats for Endangered Species at Their Historical Sites—Conditions for the Success of *Salix lapponum* and *Salix myrtilloides* Reintroduction

**DOI:** 10.3390/ijerph20021133

**Published:** 2023-01-09

**Authors:** Michał Arciszewski, Magdalena Pogorzelec, Urszula Bronowicka-Mielniczuk, Michał Niedźwiecki, Marzena Parzymies, Artur Serafin

**Affiliations:** 1Department of Hydrobiology and Protection of Ecosystems, University of Life Sciences in Lublin, Dobrzańskiego, 20-262 Lublin, Poland; 2Department of Applied Mathematics and Computer Science, University of Life Sciences in Lublin, Głęboka 28, 20-612 Lublin, Poland; 3Institute of Horticultural Production, University of Life Sciences in Lublin, Głęboka 28, 20-612 Lublin, Poland; 4Department of Environmental Engineering and Geodesy, University of Life Sciences in Lublin, Leszczyńskiego 7, 20-069 Lublin, Poland

**Keywords:** relict species, conservation, translocation, habitat conditions

## Abstract

Restoring endangered plant species to their historical sites is not always possible due to constantly changing habitat conditions. The aim of this study was to test the effects of reintroduction of two relict willow species in eastern Poland. The experiment consisted of planting 48 individuals of *Salix lapponum* and *S. myrtilloides*, obtained by micropropagation, at each of the two selected sites and observing their survival after one year. At the same time, selected physicochemical and biocenotic factors of the environment were monitored. About 70% of *S. lapponum* individuals and 50% of *S. myrtilloides* plants survived the one-year period. This result can be considered satisfactory and confirms the effectiveness of this means of active protection. The results of measurements of selected abiotic factors of the environment and the observations and ecological analysis of the flora indicated that the habitat conditions of both historical sites have changed, resulting in accelerated succession of vegetation. However, complete habitat degradation did not occur, although the development of a multi-story structure of one of the phytocenoses intensified competition for light and other environmental resources, which narrowed the potential ecological niche of the reintroduced species.

## 1. Introduction

In the era of mass extinction of species, impoverishment of biodiversity, and the climate crisis, there is an urgent need for action to protect the greatest possible number of species and entire ecosystems all over the world [1]. One tool that is increasingly exploited for nature conservation is species translocation, which involves planned and deliberate transfer of plant or animal specimens to other sites in order to increase the chances of restoration of the population or even the survival of the species. Despite the growing popularity of these measures, the ‘experiments’ carried out do not always succeed, due to multiple factors that can influence the outcome of the translocation process [2].

For the translocation process to be effective, it must be grounded in scientific data [3]. Even minor gaps in knowledge of the biology of a given species or its habitat requirements pose a risk to the success of the procedures [4]. Active dialogue on the success or failure of this type of action would undoubtedly increase the effectiveness of species translocation as a means of conservation. However, little information has been published on failures of reintroduction, while according to Abeli and Dixon [3], this could provide a great deal of valuable information helping to avoid committing the same errors in the future. The authors also claim that greater emphasis should be placed on the consideration of ecological relationships in translocation procedures. 

Climate change results in shifts in the natural ranges of occurrence of many species, but this process is often delayed in relation to the progressive effects of global warming. This is because some taxa are unable to actively overcome anthropogenic barriers, which results in isolation of their local populations [5]. Endemic and relict species are particularly sensitive to environmental changes. Their inability to effectively disperse, in combination with their narrowing ecological niche, is a real threat which can lead to the extinction of various taxa [6,7]. Alpine boreal relicts currently found at lowland sites are exposed to numerous stress factors associated with climate change and human interference in natural ecosystems. Their survival in refuges largely depends on their adaptability, which stems from their capacity for sexual reproduction. On the other hand, for species with specific habitat preferences, the stability of the conditions in which they exist is of enormous importance [8]. However, endangered species have a chance to survive if there are suitable surrogate habitats outside the model region of their occurrence. Thus, contemporary nature conservation should focus on management of populations in the context of progressive transformations of the environment, irrespective of the possibility of restoring the entire plant community of which they are a characteristic element [9]. The example of two species from the Iberian Peninsula, characterized by sparse populations (*Rhododendron ponticum* ssp. baeticum and *Impatiens glandulifera*), can be cited here. Following the deliberate introduction of these species north into Britain, they have now become common, widespread species in the area. It is difficult to estimate how many currently threatened species could survive climate change if they were relocated in time to other sites. However, the known examples of successful translocations of species give hope for the success of such efforts in the future [9].

The many widespread and well-known willow species in Poland include extremely valuable glacial relicts: the downy willow *Salix lapponum* (L.) and the swamp willow *Salix myrtilloides* (L.). These plants are associated with specific habitats found mainly in peatlands. According to the IUCN (International Union for Conservation of Nature) classification, *S. lapponum* is an endangered species (EN). It has a similar status in several European countries, such as the Czech Republic, Ukraine and Lithuania, where there are few sites of this species [10]. The swamp willow is a critically threatened plant in the Czech Republic and Slovakia [11]. In Germany it is threatened with extinction, in Ukraine it is vulnerable, and in Lithuania it is described as a rare plant [12]. In Poland, both species are strictly protected and have the status of endangered species due to the substantial decline in the number of sites and the size of their populations in recent years [12,13]. 

The natural range of these two species covers the Arctic Circle in northern Eurasia and the Scandinavian Peninsula, where the plants are widespread [10]. Outside the center of its range, *S. lapponum* is found in the British Isles, the Sudeten Mountains, Bulgaria, France, and the Pyrenees [13]. In Central Europe a few sites of isolated populations of *S. myrtilloides* are known in Alpine peatlands, the Carpathian Mountains, and the Sikhote-Alin mountain range in Russia [14].

*S. lapponum* is a small shrub growing to a height of one meter. Its characteristic feature is its elliptical, downy, greyish-green leaves with a silver-grey underside [13,15]. *S. lapponum* prefers sunny or partially shaded sites with a meso- or oligotrophic acidic (pH 4–6) substrate—bog soil and peat rich in organic matter [16]. *S. myrtilloides* is a small shrub growing to a height from 0.5 to 1 m. The aerial shoots are thin and brittle with a greenish-brown color. The leaves are initially oval, but sharply pointed in adult specimens [12,15]. The swamp willow grows best in sites not covered by shrubs. It prefers a peat substrate with a low reaction (pH 3.5–5.5). Its optimum biocenotic conditions are sedge communities of the class *Scheuzcherio-Caricetea fuscae* [12].

One of the countries in which measures have been undertaken to learn the specifics of the functioning of these species and their active conservation is Scotland. In 2002–2005 and in 2022 studies were carried out, including analyses of genetic variation, to identify threats to the small populations of *S. lapponum* in mountain ecosystems [17,18]. This species has also been the subject of ecological research conducted in the Giant Mountains (Czech Republic). The distribution and status of those isolated populations were established, and the habitat characteristics at their sites were characterized [19,20,21]. Most information on *S. lapponum* and *S. myrtilloides* comes from Poland, where studies on the ecology and biology of these species have been conducted since the 1950s [22,23,24,25,26,27,28,29,30]. Attempts have also been made to restore populations of these species to the sites of their previous occurrence and to replenish their existing populations in eastern Poland [31,32,33].

In 2021–2022 an experiment was conducted involving the reintroduction of two relict willow species, *S. lapponum* and *S. myrtilloides*, in locations in eastern Poland with extremely different habitat conditions (a raised bog and a transitional bog). Large populations of these plants had been observed at both sites in the past [34]. On the assumption that these species have a wide spectrum of ecological tolerance for the physicochemical factors typical of peatland habitats [29,30], an attempt was made to answer the following questions: Is it worthwhile to rebuild populations of these species at their historical sites, whose habitats are characterized by a high degree of naturalness but are undergoing changes caused in part by ecological succession? Does the effectiveness of reintroduction procedures, measured as the number of individuals that survive and begin their sexual reproduction cycle, confirm or call into question the sense of active conservation measures involving the translocation of plants obtained in ex situ cultivation?

## 2. Materials and Methods

### 2.1. Steps Taken before the Experiment

One of the first stages of the attempt to rebuild and replenish *S. lapponum* and *S. myrtilloides* populations was to obtain plant material. In the case of species whose populations are endangered and small, one way to obtain the greatest possible number of new plants is by in vitro micropropagation [35,36,37]. To this end, pieces of shoots 5 cm in length were collected from the largest natural populations of both willow species in eastern Poland. To maintain high genetic diversity in the new populations, the material was taken from numerous individual plants. The explants were propagated in vitro using appropriately selected media. The cultivation conditions were specific for the species, ensuring optimal growth and protection against pathogens. Following spontaneous rooting of plants in the medium, they were transferred to a substrate of peat, sand and perlite and placed in special greenhouses, where they underwent a gradual hardening process for about two months. When the plants had attained a height of 25 cm they were pruned several times to stimulate the production of lateral shoots. After a few months, plants with a height of about 40–50 cm were obtained, with fully developed leaves and woody shoots [14,38]. These plants underwent acclimation at a field station located a short distance from the border of Poleski National Park, where the weather conditions prevailing in their natural habitats could be simulated. After four weeks the plants were ready to be transplanted into peatland ecosystems [31,32].

### 2.2. Reintroduction Experiment

Two locations in Poleski National Park (eastern Poland) were selected for the reintroduction of *S. lapponum* and *S. myrtilloides* in 2021. The last documented data on the presence of these species in the plant communities of both sites date back to 1958 [22,23]. The two sites were chosen to differ in the type of peatland and degree of advancement of ecological succession. The first was located in a transitional bog where the phytocenosis had a varied species and spatial structure, the Spławy complex (S), while the other was situated on a raised bog with highly natural flora in the peatland complex Durne Bagno (DB) (Figure 1).

Plants transported in pots from the acclimation station were planted in 2021, by placing them together with the entire root ball in a floating mat. An experimental plot measuring 10 × 10 m was marked out at each site, and 48 plants of each species, *S. lapponum* and *S. myrtilloides*, were planted in each plot. At both sites both male and female individuals of both species were planted, from different clone lines obtained in vitro.

To determine the success of reintroduction, after one year the size of the new populations was determined at both sites, as well as the percentage of flowering plants among those that had survived to the next stage of vegetation.

### 2.3. Assessment of Habitat Conditions at the Reintroduction Sites

An important element of the entire species reintroduction process is a thorough analysis of the biocenotic and abiotic conditions in the habitats to which the plants will be transferred and in which they will grow following transplantation [31]. From May to September 2021, at two-week intervals, in situ measurements of the acrotelm water were made at the sites of reintroduction of the species (the YSI 556 MPS multiparameter meter was used to measure temperature, pH, electrolytic conductivity, and dissolved oxygen content). Samples of groundwater (1 dm^3^) were also collected from soil piezometers (perforated PVC pipes 1 m in length and 10 cm in diameter, closed with a plug) previously installed in both experimental plots. A total of 22 groundwater samples—11 from each site—were collected. Laboratory analyses of the water were performed to determine the values of selected physicochemical parameters of the habitats. Standard methods were used to determine the content of total nitrogen N_tot_ (photometric detection of generated nitrate after UV and thermos digestion via reduction to nitrite and azo dye formation; flow analysis method according to DIN EN ISO 29441), nitrites N-NO_3_ and nitrates N-NO_2_ (photometric detection via azo dye formation; flow analysis method according to DIN EN ISO 13395), and ammonium nitrogen N-NH_4_ (photometric detection by gas diffusion and color indicator; flow analysis method according to ISO 11732). The concentration of total phosphorus P_tot_ and phosphates P-PO_4_ was determined by spectrophotometry with ammonium molybdate, and total organic carbon (DOC) was determined using an automatic PASTEL UV analyzer.

Statistical analysis of the results of the measurements of the physicochemical parameters of the water was performed. The distribution of parameters was presented graphically in boxplots. The nonparametric Mann–Whitney U test was used to compare the distribution of the physicochemical parameters of the water at the two sites, for a significance level of 0.05. Principal component analysis (PCA) was performed to analyze the structure of the data (PCA). Computations and visualization were performed in Statistica 13 software in the R environment [39] ver. 4.1.0.

For a characterization of biocenotic conditions (differences in vegetation in the two habitats), species of vascular flora were listed. To take into account seasonal changes in vegetation, phytosociological relevés were made twice (in July 2021 and May 2022) in the experimental plots [40]. Based on the results, an attempt was made to classify the plant communities to appropriate syntaxonomic units according to Matuszkiewicz et al. [41].

The phytocenoses of the experimental plots were analyzed to determine the habitat preferences of individual species, using ecological indicator values of vascular plants [42].

## 3. Results and Discussion

Reports of translocation of endangered species as a means of active conservation are increasingly appearing in the literature [43,44,45]. The success of these efforts is determined by multiple factors affecting various stages of the process. Their foundation is in-depth knowledge of the biology and ecology of the species and verification of population resources, potential threats, and their sources [46]. The source of the plant material, as well as its form (seeds or vegetative parts of plants), unquestionably has an important influence on the success of measures aimed at restoring valuable plant taxa to their natural habitats. The parent plants from which the material for the formation of new populations is obtained should come from more than one source or from a stable population with high genetic diversity [47,48]. Another decisive factor for the long-term success of reintroduction is the choice of a suitable habitat in accordance with the preferences of the species. An important challenge, however, is to predict the changes that may take place in the ecosystems as well as in the microhabitats of the surrogate sites. Therefore, habitats must be carefully selected to reduce the risk of translocation failure [4,31].

Literature data on successful reintroductions is much more common than information on failures. Godefroid et al. analyzed 249 documented translocations around the world, focusing on assessment of the methods used and their outcomes. As a result, they distinguished three main indicators of the effectiveness of plant reintroduction procedures and found that the average survival rate of the individuals introduced to the environment was 52%, with a flowering rate of 19% and a fruiting rate of 16%. These values were found to decrease over time [4].

In eastern Poland, efforts undertaken for several years have resulted in successful reintroduction of *S. lapponum* and *S. myrtilloides*. Researchers describe the results of monitoring of *S. lapponum* populations following translocation of plants in 2018 and 2019 to sites in the forest bog Blizionki, where the species had once been present. The effectiveness of reintroduction, measured as the number of individuals that survived a year in the new habitat, was estimated at 66% and 80% in successive years [31,32,49]. Flower formation by reintroduced plants is one of the most important indicators of their condition and predictors of the success of conservation procedures [4]. Flowering rates in the described group of plants ranged from 15% to 18% for the plants introduced at the surrogate sites [31].

Although the ultimate success of reintroduction cannot usually be judged until at least several years have passed, the high survival rate of the plants one year after planting may be a promising sign that the goal will be achieved [31].

Evaluation of the new *S. lapponum* and *S. myrtilloides* populations created in the experiment, carried out a year after planting, suggests the need for careful selection of habitats, given the pronounced disproportions in the numbers of plants that survived in the natural habitat. In the case of *S. lapponum*, while at site S (the transitional bog habitat) a substantial number of individuals survived, i.e., 67% of those planted a year earlier (of which 30% flowered), at site DB only 27% survived (of which 23% produced inflorescences). The reverse pattern was observed for *S. myrtilloides*, as more reintroduced plants survived on the raised bog at site DB (50%, of which 30% flowered), but only 16% at site S (of which 30% began to flower) (Figure 2).

The substantial differences between the surrogate sites where reintroduction was carried out in 2021 are confirmed by the results of the in situ measurements of selected physicochemical parameters of the environment and by the laboratory analyses of water sampled from the sites.

Due to the lack of compliance with the normal distribution, the Mann–Whitney U test was used to compare the parameters of the water sampled from the bogs (experimental plots S and DB). Significant differences were obtained in the distribution of the following parameters: pH (*p* < 0.001), electrolytic conductivity (EC) (*p* < 0.001), and oxygen saturation of the water from the two experimental plots (*p* < 0.001), TOC (*p* < 0.001), NO_2_ (*p* < 0.001) and NH_4_ (*p* < 0.05). No significant differences (*p* > 0.05) were found for the remaining parameters (N_tot_, P_tot_, N-NO_3_, P-PO_4_, and water temperature).

The first parameter, most strongly differentiating the habitat conditions of the sites of reintroduction of relict willows, was the reaction of the acrotelm water (Figure 3). At the Durne Bagno bog (experimental plot DB), it ranged from pH 3.7 to 4.3. The water reaction measured at site S (Spławy) underwent less variation over the seven months of measurements, ranging from pH 6.0 to 6.2 (Figure 3). Interestingly, the degree of success of reintroduction was not what might be expected based on the habitat preferences of *S. lapponum* and *S. myrtilloides*. According to Zarzycki et al. [42], *S. lapponum* prefers lower values for substrate reaction (pH 4.0–5.0) than *S. myrtilloides* (pH 4.0–6.0), but more individuals of these species survived in the conditions of experimental plot S, where the reaction during the entire growing period was pH > 6.0. In the case of *S. myrtilloides*, based on the habitat preferences of the species, reintroduction was expected to be more effective at the site in the transitional bog (S). Evaluation of the population, however, revealed a different outcome, which may mean that substrate acidity is not a habitat factor limiting the growth and survival of either *S. myrtilloides* or *S. lapponum*. However, it does indicate that the species have a relatively wide range of ecological tolerance, as confirmed in previous research. At lowland sites, the water reaction of the habitats with a functioning population of *S. lapponum* ranged from pH 5.1 to 6.7 [29]. Studies conducted on mountain populations (the Giant Mountains in the Czech Republic) showed that it grows in soil with pH 4.2–4.7 [20]. In the case of *S. myrtilloides,* results ranged from about pH 4.7 to 6.2, depending on the location [30].

Another parameter indicating differences between the habitats at the experimental sites was the electrolytic conductivity of the water. The values for this parameter were lower in the case of samples collected in 2021 from the Durne Bagno bog (from 60 to 190 μS/cm on various days) than at the site in the Spławy bog, where the values ranged from 250 to 410 μS/cm (Figure 3). Relatively high dissolved oxygen content was noted in the water samples from the Spławy bog (average 150 mg/L, max 200 mg/L in late July/early August 2022). In contrast, at no time did the value of this parameter in the water from the Durne Bagno bog exceed 90 mg O_2_/L (most often ranging from 40 to 50 mg/L) (Figure 3).

The values of the remaining physicochemical parameters tested in the water of the habitats were similar. The Spławy and Durne Bagno bogs did not differ substantially in the content of phosphorus or nitrogen fractions in the acrotelm water. The content of total phosphorus and phosphorus in the form of orthophosphate ions in the water was low. The average values were 0.54 mg P-PO_4_/L and 0.75 mg P_tot_/L in the water samples from experimental site DB, and 0.17 mg P-PO_4_/L and 0.37 mg P_tot_/L for the water from site S (Figure 3). Literature data confirm that the test species prefer oligotrophic habitats or moderately poor mesotrophic habitats [42]. The low content of nitrogen and phosphorus fractions in the water of the habitats of the *S. lapponum* and *S. myrtilloides* populations in eastern Poland is also described in literature [29,30]. The results of research carried out in 2011–2013 were comparable to those characterizing the habitats of the experimental plots (in 2021), as the average phosphorus content in the water ranged from 0.1 to 0.6 mg/L.

In addition to the complex physicochemical factors complementing one another in the habitat, a crucial role is played by the relative stability of their values over time, which guarantees stable conditions for organisms. Statistical analysis of the tests carried out in 2021 made it possible to track the changes in the values of selected abiotic environmental factors between successive measurements at the two experimental plots. The structure of the data was evaluated using principal component analysis (PCA), in which all measurement times were included as cases, and the parameters pH, EC, oxygen saturation, N_tot_ and P_tot_ were variables. On the basis of the PCA, two main components were distinguished, which explain 80.08% of the overall variability of the data set (Figure 4). The first component (PCA1) explains 61.64% of the total variation, and the second component (PCA2) explains 18.44% of the total variation. The results are presented in graphic form in Figure 4, where it can be seen that the results of measurements of environmental factors obtained at different times in the Durne Bagno bog were more varied than at the Spławy bog, and that the values of the parameters of the water sampled in the summer (late July/early August) differ noticeably from those determined in spring and autumn. A similar pattern can be seen for the data collected at the site in the Spławy bog. The results of the measurements can also be seen to differ markedly between the two habitats, especially in summer (Figure 4a). Note that variables such as pH, EC and %O_2_ are strongly correlated with the negative part of the first component (PCA 1), while the other variables, i.e., N_tot_ and P_tot_, are correlated with the negative part of the second component (PCA2) (Figure 4b).

The variability of abiotic habitat factors has a pronounced influence on the living part of the ecosystem. In this context, changes in the spatial and species structure of the flora are particularly evident. This is also true of the experimental plots where specimens of the two relict species were reintroduced in 2021.

Historical data from the 1950s regarding sites of *S. myrtilloides* in the Łęczna-Włodawa Lakeland (eastern Poland), published by Fijałkowski, indicate that the species was mainly present in transitional bogs [23]. All plant communities in which its presence was recorded belonged to the class *Scheuchzerio-Caricetea fuscae* and the orders *Caricetalia fuscae* and *Sphagnetalia.* The most common plant associations that included *S. myrtilloides* were *Caricetum diandre*, *Caricetum lasiocarpae* and *Carex limosa-Scheuchzeria palustris.* Plant components common to the phytocenoses studied by Fijałkowski were *Salix cienera*, *Betula humilis*, *Betula pubescens*, *Menyanthes trifoliata*, and *Carex lasiocarpa*. The author also emphasized that *S. myrtilloides* was often accompanied by *S. lapponum* in the habitats. On the other hand, the results of research conducted more than half a century later in this area suggest that biocenotic changes had taken place in the ecosystems, as in 2002 and 2012 plants of the classes *Oxycocco-Sphagnetea*, *Alnetea glutionosae* and *Scheuchzerio-Caricetea* and the alliances *Magnocaricion* and *Alno-Ulmion* were dominant [27].

The plant communities of experimental plots S and DB differed substantially in physiognomy and species composition, as confirmed by the phytosociological relevés made in 2021 and 2022. The first visible difference was the percentage of trees and shrubs in the phytocenosis and their species composition (Table 1). In experimental plot S there was a large percentage of species of the families Betulaceae and Salicaceae: *Alnus glutinosa*, *Betula pubescens*, *Betula humilis*, *Salix cienera* and *Salix pentandra*. In some places, individuals of these species formed dense clusters. In the herb level, a number of vascular plant species were identified, among which *Oxycoccus palustris*, *Comarum palustre* and *Thelypteris palustris* were dominant. *Carex limosa* and *Typha angustifolia* covered parts of the area. The presence of endangered species *Drosera rotundifolia* and *Menyanthes trifoliata* was noted as well. The lowest layer of the community was formed by various species of *Sphagnum* spp. mosses. The phytocenosis of the experimental plot located in the Spławy (S) bog was syntaxonomically highly diverse. Woody species characteristic of the class *Alnetea glutinosae* were recorded, with dominance of *Alnus glutinosa*, while the remaining plant species belonged to the classes *Scheuchzerio-Caricetea nigrae* and *Oxycocco-Sphagnetea.* Analysis of the phytosociological relevés made it possible to classify the phytocenosis as the association *Betulo-Salicetum repentis*, i.e., a group of scrub communities. A characteristic feature of this association is the substantial share of species of the class *Scheuchzerio-Caricetea nigrae* and the pronounced boreal continental nature of the community [41].

The Durne Bagno bog had a distinctive structure of wet depressions alternating with hummocks of drier peat moss and relatively low species richness of flora (Table 1). In the shrub layer of experimental site DB, isolated individuals of *Pinus sylvestris* and *Betula pubescens* were noted. The species with the largest share of the herb layer were *Eriophorum vaginatum* and *Oxycoccus palustris*. *Rhododendron tomentosum* and *Juncus conglomeratus* were observed in the part of the plot located nearest the edge of the bog. The vegetation colonizing the plot in the Durne Bagno bog was characteristic of the class *Oxycocco-Sphagnetea*—communities of small shrubs and *Sphagnum* typical of wet heaths and high bogs with an acidic pH, with water supplied mainly by precipitation. Based on the high proportions of *Eriophorum vaginatum* and *Oxycoccus palustris* forming distinct clusters, it was classified as the order *O. Sphagnetalia magellanici* [41].

Analysis of the preferences of the flora species (according to Zarzycki et al.) [42] of both habitats (S and DB) reveals that most of the plants there function normally at sites with moderate light, and only a small proportion of the plants require fully lit sites. At both the Spławy bog and the Durne Bagno bog, species requiring wet soil rich in organic matter made up the largest share of the communities (about 70–80%). However, pronounced differences were noted in the soil fertility preferences of species. At site S, more than 50% of the species observed in the phytocenosis were characteristic of poor and moderately poor soil, while about 46% of the flora there comprised species preferring nutrient-rich (eutrophic) soil. The vegetation of site DB was not so diverse in this respect, consisting mainly of species that develop on mesotrophic, oligotrophic and even extremely nutrient-poor soils, preferring an acidic (4 ≤ pH < 5) or moderately acidic (5 ≤ pH < 6) substrate. In contrast, the flora of the Spławy bog (S) was represented by plants that require a neutral (6 ≤ pH < 7) or even alkaline (pH < 7) substrate reaction. However, the range of requirements of the species present at experimental site S was wide, and plants preferring acidic and moderately acidic soil were recorded there as well (Table 2).

The differences in the physiognomy of the phytocenoses (at experimental sites S and DB) and the habitat requirements of the species forming them draw attention to the process of ecological succession. Its accelerated course at the Spławy bog (S) is first of all indicated by the species richness of the flora, with highly diverse habitat preferences (Table 2). The species composition also included a substantial proportion of woody species and potentially expansive species belonging to communities of the class *Phragmitetea*, posing a threat of competition for light and other resources for the reintroduced plants. This may have been the reason for the less successful than expected reintroduction of *S. myrtilloides*, which, for comparison, survived in much greater numbers than *S. lapponum* at the Durne Bagno bog, where competition from other woody species was very low.

## 4. Conclusions

Based on the results of measurements of selected abiotic environmental factors and the data obtained during observations and ecological analysis of the flora of the sites of experimental reintroduction, it cannot be conclusively determined what had the decisive influence on the outcome of the procedures. The results of monitoring of the new populations of *S. lapponum* and *S. myrtilloides* a year after their transfer to the sites revealed pronounced disproportions in the numbers of plants that survived for 12 months in the surrogate habitats. It appears likely, however, that the conditions in the raised bog (Durne Bagno) were more favorable to plants of the species *S. myrtilloides*, while the structure of the phytocenosis and the set of abiotic factors at the transitional bog (Spławy) did not substantially limit the growth and development of *S. lapponum*. The habitat conditions at both sites, which had once been colonized by populations of these species, have undergone changes resulting in accelerated succession of vegetation. However, they have not been completely degraded, although development of the multi-story structure of the phytocenosis of the Spławy bog increases competition for light and other environmental resources, thus narrowing the potential ecological niche of the reintroduced species.

Reintroduction resulted in the survival of no more than 70% of the *S. lapponum* plants and 50% of the *S. myrtilloides* plants. Given the experimental nature of introduction of plants to habitats with extremely different abiotic and biocenotic conditions, this outcome can be considered to be satisfactory and to confirm that active conservation by means of translocation of plants cultivated ex situ is possible and justified.

## Figures and Tables

**Figure 1 ijerph-20-01133-f001:**
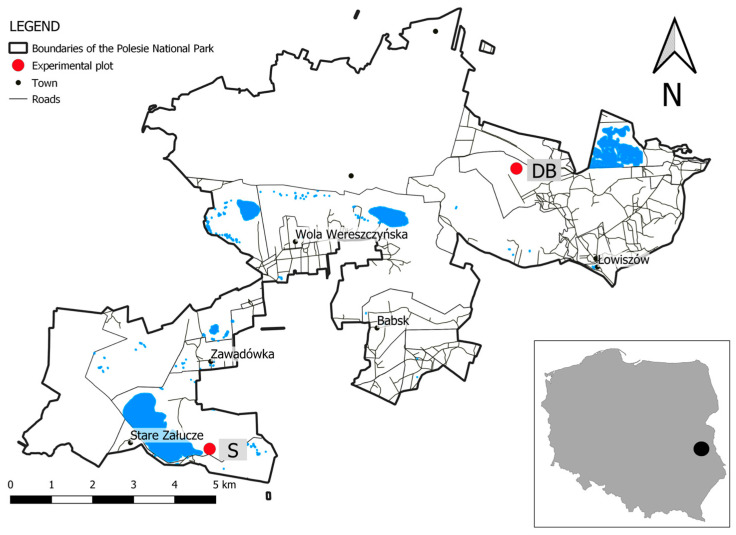
Location of experimental plots: DB—Durne Bagno (51°27′05.83″ N/23°13′33.29″ E), S—Spławy (51°24′19.96″ N/23°06′16.99″ E).

**Figure 2 ijerph-20-01133-f002:**
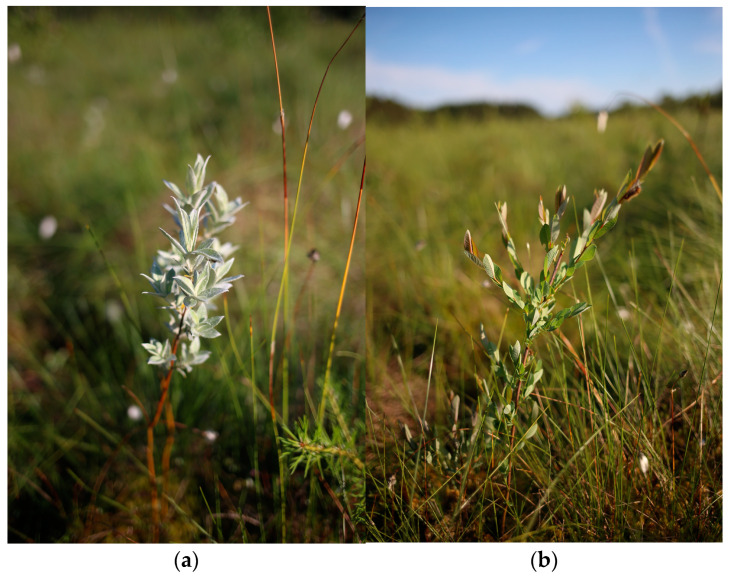
Plant individuals in newly established populations (**a**) *S. lapponum*; (**b**) *S. myrtilloides*.

**Figure 3 ijerph-20-01133-f003:**
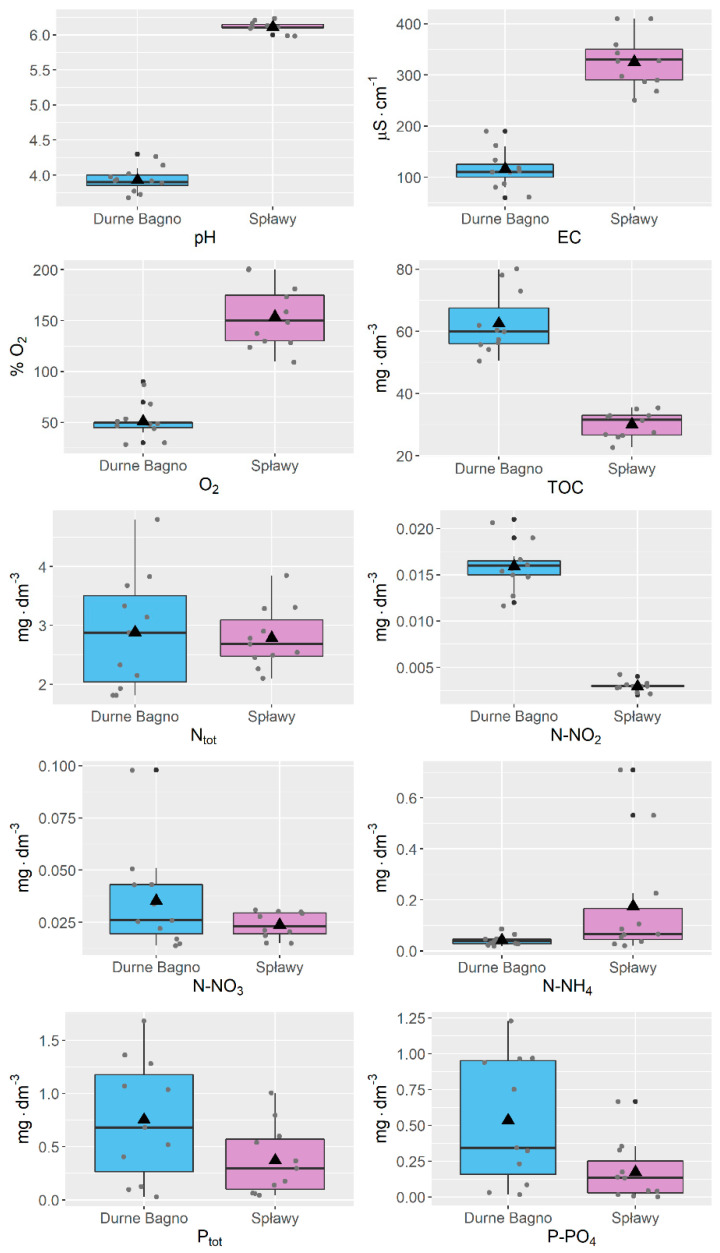
Distribution of physicochemical parameters at the sites of reintroduction of relict willows. Data from 11 measurements made from May to September 2021. The horizontal line across the central region of the box represents the median. The mean value of the data is marked by a filled triangle. The whiskers are drawn to the most extreme observations. Any observation not included between the whiskers is plotted as an outlier with a filled dot. The plot presents observed values of parameters, marked with small grey dots.

**Figure 4 ijerph-20-01133-f004:**
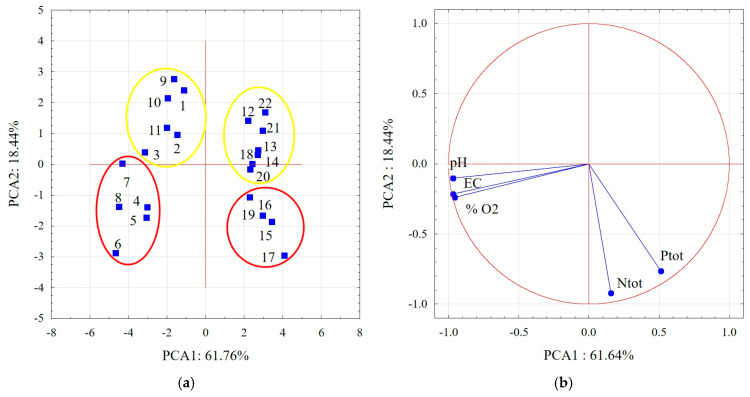
Results of principal component analysis for (**a**) individual measurement times at the study sites, where 1–11 and 12–22 denote successive measurements of physicochemical factors of the water at experimental sites DB and S, respectively; the color red indicates measurements made in summer; yellow denotes measurements in spring and autumn; (**b**) physicochemical parameters of the water.

**Table 1 ijerph-20-01133-t001:** Species composition of the phytocenoses of the experimental plots (Spławy and Durne Bagno) in 2021/2022, taking into account the characteristic cover abundance of individual vascular plant species (Braun-Blanquet’s scale).

Vegetation Stratum	Species	Experimental Plot S	Experimental Plot DB
B	*Alnus glutinosa* L.	3	
	*Betula humilis* Schrank.	2	
	*Betula pubescens* Ehrh.	3	1
	*Pinus sylvestris* L.		1
	*Salix cinerea* L.	2	
	*Salix pentandra* L.	2	
C	*Carex limosa* L.	2	2
	*Cirsium palustre* L.	2	
	*Comarum palustre* L.	3	
	*Drosera rotundifolia* L.	1	
	*Eriophorum vaginatum* L.	+	4
	*Geranium palustre* L.	+	
	*Juncus conglomeratus* L.		2
	*Lysimachia thyrsiflora* L.	2	
	*Lythrum salicaria* L.	2	
	*Menyanthes trifoliata* L.	2	
	*Oxycoccus palustris* L.	3	4
	*Peucedanum palustre* L.	+	
	*Ranunculus acris* L.	+	
	*Rhododendron tomentosum* L.		2
	*Silene flos-cuculi* L.	+	
	*Thelypteris palustris* Schott	3	1
	*Typha angustifolia* L.	2	2
D	*Sphagnum* spp.	4	5

**Table 2 ijerph-20-01133-t002:** Proportions of ecological groups of plants in the phytocenoses of the habitats (based on ecological indicator values of vascular plants: L—light value; W—soil moisture value; Tr—trophy value, R—water acidity value, H—organic matter content value; numerical designations denote assignment to specific ecological groups following Zarzycki et al., 2002 [42]).

Experimental Plot	Indicator Value	L (%)	W (%)	Tr (%)	R (%)	H(%)
Spławy (S)	1	-	-	4.42	4.17	1.50
	2	1.81	-	12.71	13.54	10.53
	3	10.9	1.07	36.5	18.95	87.97 *
	4	87.2 *	22.78	46.41 *	52.63 *	-
	5	-	67.62 *	-	10.53	-
	6	-	8.54	-	-	-
Durne Bagno (DB)	1	-	-	15.52	13.1	1.85
	2	2.86	1.90	51.72 *	26.19	3.70
	3	12.86	2.86	25.86	35.71 *	94.44 *
	4	77.14 *	19.05	6.90	19.05	-
	5	7.14	70.07 *	-	5.95	-
	6	-	-	-	-	-

* indicate dominant ecological groups of plants in each phytocenosis.

## Data Availability

Not applicable.

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
