# Peer review of "The Search for Suitable Habitats for Endangered Species at Their Historical Sites—Conditions for the Success of *Salix lapponum* and *Salix myrtilloides* Reintroduction"

_ijerph, 2023, doi:10.3390/ijerph20021133_

Round 1

Reviewer 1 Report

Paper looks very nice and sounds  interesting

I appreciate the authors of the paper ‘The Search for Suitable Habitats for Endangered Species at  their Historical Sites – Conditions for the Success of Salix Lapponum and Salix Myrtilloides Reintroduction’ for taking initiative in the conservation of two endangered plant species S. lapponum and S. myrtilloides by reintroducing them at their historical sites. However, few suggestions are pointed down which may improve your manuscript.

1.  The authors have mentioned about the existence of the S. lapponum and S. myrtilloides in Poland in the past years (L- 103, 138). I suggest you to add the exact year of the last occurrence of the two species within the same sites so that you can compare and analyze if the success rates of reintroduced plants varies according to the duration of their last occurrence and the reintroduction.

2. Full species name Salix lapponum and Salix myrtilloides is repeatedly used with the first under each sub-title. It is better to stick on to S. lapponum and S. myrtilloides later on after the first use of the full species name throughout the text rather than considering each sub-sections (eg: L- 117, 137, 213, 234, 423).

3. Since numbers are used to represent references within the text, it is better to provide substantive results without mentioning the first author. There are few (L- 206, 213 etc.) instances in the text where the author’s name is mentioned along with the respective results. I suggest to follow a uniform pattern by rephrasing such sentences, if possible.

Reviewer 2 Report

General comments

I read the manuscript with great interest. All the sections read well and are easy to follow. I think the manuscript can be accepted with a minor revision. The study site is located in a national park; therefore, discussing Post 2020 biodiversity framework targets would be worthwhile, especially how this study would help achieve conservation targets. Besides, the authors should cite similar studies but for other taxa.

Specific comments

From my perspective, the title is a bit long. Maybe shorten it?

Line 33: The authors could also cite literature from other taxa. Given this work was conducted in a national park, the following two references might be cited.

Chowdhury, S., Jennions, M. D., Zalucki, M. P., Maron, M., Watson, J. E., & Fuller, R. A. (2022). Protected areas and the future of insect conservation. Trends in Ecology & Evolution.

Thomas, C. D. (2011). Translocation of species, climate change, and the end of trying to recreate past ecological communities. Trends in Ecology & Evolution, 26(5), 216-221.

Figure 1. Please add all the details in the figure caption. The figure caption should always be detailed. Besides, avoid using ‘red’.
